# Perspectives on simulation-based training from paediatric healthcare providers in Nigeria: a national survey

Rachel Umoren [1,2] Veronica Chinyere Ezeaka,[3] Ireti B Fajolu,[3]
Beatrice N Ezenwa [3] Patricia Akintan,[3] Emeka Chukwu,[1] Chuck Spiekerman[4]

2019

[1]Department of Pediatrics, University of Washington School of Medicine, Seattle, Washington, USA
[2]Department of Pediatrics, Seattle Children's Hospital, Seattle, Washington, USA
[3]Department of Paediatrics, College of Medicine, University of Lagos, Lagos, Nigeria
[4]Department of Biostatistics, University of Washington, Seattle, Washington, USA

**Correspondence to**
Dr Rachel Umoren;
rumoren@uw.edu

## ABSTRACT

**Objectives** The objective of this study was to explore the access to, and perceived utility of, various simulation modalities by in-service healthcare providers in a resource-scarce setting.

**Setting** Paediatric training workshops at a national paediatric conference in Nigeria.

**Participants** All 200 healthcare workers who attended the workshop sessions were eligible to participate. A total of 161 surveys were completed (response rate 81%).

**Primary and secondary outcome measures** A paper-based 25-item cross-sectional survey on simulation-based training (SBT) was administered to a convenience sample of healthcare workers from secondary and tertiary healthcare facilities.

**Results** Respondents were mostly 31–40 years of age (79, 49%) and women (127, 79%). Consultant physicians (26, 16%) and nurses (56, 35%) were in both general (98, 61%) and subspecialty (56, 35%) practice. Most had 5–10 years of experience (62, 37%) in a tertiary care setting (72, 43%). Exposure to SBT varied by profession with physicians more likely to be exposed to manikin-based (29, 30% physicians vs 12, 19% nurses, p<0.001) or online training (7, 7% physician vs 3, 5% nurses, p<0.05). Despite perceived barriers to SBT, respondents thought that SBT should be expanded for continuing education (84, 88% physician vs 39, 63% nurses, p<0.001), teaching (73, 76% physicians vs 16, 26% nurses, p<0.001) and research (65, 68% physicians vs 14, 23% nurses, p<0.001). If facilities were available, nearly all respondents (92, 98% physicians; 52, 96% nurses) would recommend the use of online simulation for their centre.

**Conclusions** The access of healthcare workers to SBT is limited in resource-scarce settings. While acknowledging the challenges, respondents identified many areas in which SBT may be useful, including skills acquisition, skills practice and communication training. Healthcare workers were open to the use of online SBT and expressed the need to expand SBT beyond the current scope for health professional training in Nigeria.

## INTRODUCTION

A simulation is an approach to training that provides learners with an opportunity to practice their skills in a safe manner on a manikin or in a virtual space before a clinical encounter or procedure on a patient.[1 2]

### Strengths and limitations of this study

► The study was a national survey of Nigerian paediatric healthcare professionals.
► The response rate to the survey was high.
► Physicians and nurses practicing in both public and private healthcare facilities were included in the study.
► The study compared responses from health professionals working secondary and tertiary working in different parts of the country.
► As with limitations seen in other cross-sectional surveys, there is potential for selection and recall bias.

Simulation-based training (SBT) is supported by adult learning theories such as the Kolb's experiential learning theory[3 4] and the Ericsson's deliberate practice theory[5] and is near the top of the Kirkpatrick triangle for supporting increased retention of knowledge and skills.[6] For this reason, elements of SBT have been integrated into many global maternal and newborn health programme such as the Neonatal Resuscitation Program (NRP) and Helping Babies Survive.[7 8]

The majority of paediatric SBT in high-income countries is associated with standardised resuscitation training programme such as NRP and paediatric advanced life support.[7] This training is conducted in two parts using online simulation (NRP eSIM and HeartCode) and manikin-based simulation in clinical simulation facilities that are set up to mimic actual clinical settings with fixtures such as suction and gas outlets and equipment including cardiac monitors, infant warmers and hospital beds.[7] In situ simulations occur in healthcare facilities and are designed to provide convenient opportunities for practice in the healthcare setting and to identify patient safety risks.[9 10]

In low-income settings, paediatric SBT in newborn resuscitation and care using the Helping Babies Survive program is

conducted in non-clinical settings such as classrooms and hotel conference rooms with a low-cost manikin such as the Neonatalie manikin (Laerdal Medical) that can be filled with air or water and is resistant to adverse environmental conditions.[11–14] Refresher training is encouraged following initial training using manikins and resuscitation equipment at designated practice locations in healthcare facilities such as the Helping Babies Breathe Corner.[13 14]

However, there are logistical challenges to training using simulation that involve a higher teacher to student ratio and the need for simulation equipment and space in the clinical or educational setting for learners to be taught.[15–17] For these reasons, virtual simulations are increasingly considered as a complement to manikin-based training.[18 19] However, little is known of the access of healthcare providers in a resource-scarce setting towards SBT and, in particular, virtual reality (VR) simulation. The objective of this study was to explore the access to and perceived utility of various simulation modalities by in-service healthcare providers in a resource-scarce setting.

## METHODS
A 25-item cross-sectional survey was created by the investigators (RU, CE) who are simulation research collaborators from the University of Washington/Seattle Children's Hospital and the University of Lagos with questions on access to SBT facilities and perceptions on SBT in paediatric settings (see the online supplementary file). The input was obtained from experienced simulation educators and healthcare professionals practicing in the USA and in Nigeria. The survey was piloted for clarity and ease of use among Nigerian paediatric healthcare professionals and revised based on feedback. The survey was designed to be delivered in English and intended for administration to paediatric healthcare workers. The study was approved as exempt by the Seattle Children's Hospital Institutional Review Board and ethics approval in Nigeria was obtained from the University of Lagos Health Research Ethics Committee.

### Participants
The anonymous survey was administered on paper to a convenience sample of 200 healthcare workers who attended conference workshops conducted in January 2018 at the Paediatric Association of Nigeria Conference in Abuja (North Central), Nigeria. All participants were English-speaking.

### Eligibility
All workshop attendees were eligible to participate in the study and were provided with a copy of the paper-based survey, which included information about the study.

### Patient and public involvement
As this was a study of healthcare providers, patients were not involved.

## Measures
### Access to SBT facilities
Respondents were asked two questions on their access to SBT facilities: 'Does your institution/health facility have facilities for SBT' and 'Does your centre have a skills-based simulation lab?' Respondents were asked, 'In what capacity does your institution use SBT?' Respondents could select from three options that were not mutually exclusive: teaching, research or examination.

### Exposure to SBT
Respondents were asked about their awareness of and exposure to SBT modalities including manikin-based, online and VR simulation. Within the exposure domain, no examples of VR simulations specific to paediatric training were available at the time of the survey, but respondents were asked if they had ever used VR simulation.

### Challenges to SBT
Respondents were asked questions on the challenges of having a skills-based simulation lab at their centre and the challenges to online (computer-based or VR) simulation. Response options on the challenges to having a skills-based simulation lab were lack of funding, lack of access to equipment, lack of curriculum, lack of space, lack of instructors trained in simulation education and lack of awareness of an option for SBT.

### Perceptions of SBT
Respondents were asked to identify the advantages of SBT that they were aware of, whether SBT could be expanded beyond the current scope and in what way SBT should be expanded. Finally, respondents were asked whether if all facilities were available, they would recommend online simulation for their centre with response options: yes or no.

### Data analysis
Data were analysed using descriptive statistics, Pearson's $\chi^2$ test and the Fisher exact test to examine the relationship between demographic characteristics and respondents' access and exposure to SBT facilities in their institution or healthcare facility as well as their perceptions of the benefits and challenges in using SBT in their facility. We specifically compared the impact of demographic characteristics such as profession (physician or nurse), years in practice and type and location of practice; on access to SBT, perceived challenges of SBT and perceived utility of SBT. In some cases, subcategories of the profession (eg, consultant physician, registrar, house officer and medical officer), years in practice and geographic location (north vs south geopolitical zones) were collapsed for comparison due to small numbers of respondents in individual categories. No power calculation or sample size calculation was performed as the sample size was fixed, that is, healthcare workers attending the conference. SAS V.9.4 software was used for the analysis.

## Table 1  Demographics of respondents

| Demographic characteristics, n=161 | | N (%) |
|---|---|---|
| Age range (years) | 21–30 | 26 (16) |
| | 31–40 | 79 (48) |
| | 41–50 | 44 (27) |
| | >50 | 17 (10) |
| Gender | Male | 34 (21) |
| | Female | 127 (79) |
| Profession | Physician | |
| | Consultant | 26 (15) |
| | Registrar/House Officer | 45 (28) |
| | Medical Officer | 26 (16) |
| | Non-physician | |
| | Nurse/nurse-midwife | 62 (39) |
| | Community Health Extension Worker/Officer | 9 (6) |
| Years of practice | <5 | 28 (17) |
| | 5–10 | 62 (37) |
| | 11–15 | 35 (21) |
| | 16–20 | 20 (12) |
| | >20 | 21 (13) |
| Location of practice | North East | 2 (1) |
| | North West | 7 (4) |
| | North Central* | 100 (60) |
| | South East | 12 (7) |
| | South West | 32 (19) |
| | South South | 14 (8) |
| Type of healthcare facility | Government—Tertiary care | 72 (43) |
| | Government—Secondary care | 34 (20) |
| | Government—Primary care | 20 (12) |
| | Private | 41 (25) |
| Specialty | General paediatrics | 98 (64) |
| | Subspecialty paediatrics | 22 (14) |
| | Other specialties | 34 (22) |

*North Central: Abuja Federal Capital Territory (FCT), the capital city of Nigeria, is located in North-Central Nigeria and was the location of the conference.

## Table 2  Access to simulation-based training in health facilities

| Respondent characteristics (n=155) | Facilities available for simulation-based training, n (%) | P value |
|---|---|---|
| Profession | | NS |
| Physician | 62 (66) | |
| Nurse | 37 (61) | |
| Years in practice | | NS |
| >10 | 44 (62) | |
| ≤10 | 54 (66) | |
| Type of facility | | NS |
| Government | 70 (61) | |
| Private | 28 (70) | |
| Geographic location of practice | | NS |
| North | 59 (61) | |
| South | 39 (68) | |

North = North-East, North-Central, North-West Nigeria geopolitical zones.
South = South-West, South-East, South-South Nigeria geopolitical zones.

### Type and location of the practice

Respondents were mostly in general practice (98, 64%) with fewer in subspecialty paediatrics (22, 14%). Most respondents had practiced for ≤10 years (90, 54%) and many practiced in a tertiary care setting (72, 43%). The majority of respondents practice in the North Central (100, 60%) or South West parts of Nigeria (32, 19%).

### Access to SBT facilities

Table 2 shows the distribution of respondents with SBT facilities at their facility by profession, years in practice, type and location of the practice. There were no differences in access to SBT. Comparatively fewer respondents reported having a skills-based simulation lab at their centre (22, 23% physicians vs 21, 34% nurses, p=0.120).

### Exposure to SBT

Where facilities were available for SBT, most physicians and nurses reported the use of simulation facilities for teaching (physicians 62, 65%; nurses 34, 55%). There was low reported use for research (physicians 6, 6%; nurses 10, 16%) and examination purposes (physicians 21, 22%; nurses 6, 10%). Manikin-based training was more frequently reported than online simulation. The most reported type of training was basic life support (physicians 36, 38%; nurses 18, 29%). Exposure to manikin-based training varied by type of facility and geographic location (table 3).

Physicians were the group most likely to have been exposed to manikin-based paediatric training programme such as Helping Babies Breathe (29, 30% physicians vs 12, 19% nurses vs 1, 11% community health workers,

## RESULTS

A total of 161 surveys were completed (response rate 81%). Table 1 provides the demographic characteristics of respondents. The majority of respondents were under 40 years of age (105, 65%). Approximately one-third of respondents were nurses or nurses/midwives. There was a higher percentage of women represented (127, 79%) which is expected given the known predominance of women in the paediatrics and nursing professions.[20 21]

**Table 3** Exposure to manikin-based training in basic life support varies by type and location of facility

| Basic life support Manikin-based training (n=158) | N (%) | P value |
|---|---|---|
| Profession | | NS |
| Physician (consultant or registrar) | 36 (38) | |
| Nurse/nurse-midwife | 18 (29) | |
| Years in practice | | NS |
| >10 | 24 (32) | |
| <10 | 30 (33) | |
| Type of facility | | <0.001 |
| Government | 30 (36) | |
| Private | 23 (58) | |
| Geographic location | | <0.01 |
| North | 25 (25) | |
| South | 28 (48) | |

North = North-East, North-Central, North-West Nigeria geopolitical zones.
South = South-West, South-East, South-South Nigeria geopolitical zones.

p<0.001) or online training in neonatal resuscitation using the NRP eSIM (7, 7% physician vs 3, 5% nurses vs 0, 0% community healthcare workers p<0.05). Although the majority of physicians (91, 96%) and nurses (41, 72%) owned smartphones, and many were aware that VR simulations could be run on their personal phone (43, 47% physician vs 28, 51% nurses), only 3% (n=5) of all respondents had experienced a VR simulation.

## Challenges to SBT

Respondents identified challenges to having a skills-based simulation lab and to online (computer-based or VR) simulation. The lack of curriculum and lack of funding were perceived as less of a barrier to establishing a skills-based simulation lab by respondents from private

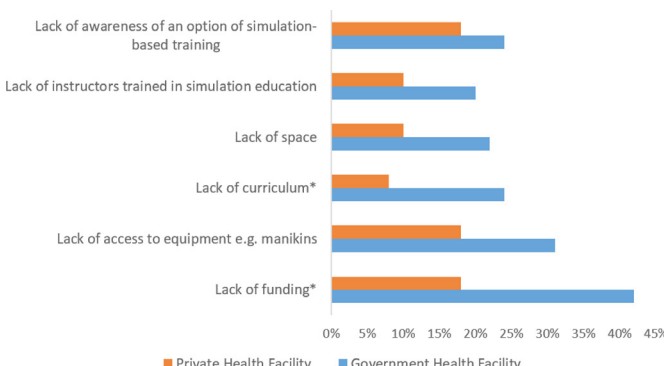

**Figure 1** Challenges to establishing skills-based simulation labs in private and government health facilities legend: *p<0.05.

healthcare facilities compared with respondents from government facilities (p<0.05) (figure 1).

The lack of awareness was the most reported challenge to using online simulation (82, 51%). Other perceived challenges to online simulation were a lack of VR equipment (37, 23%) and lack of standardised VR training modules (35, 22%). Fewer respondents reported a lack of internet access (24, 15%) or inconsistent power supply (21, 13%) as a challenge to online training.

## Perceptions of SBT

Respondents identified the advantages of SBT to include skills acquisition, provides feedback, step down training, monitoring and evaluation, debriefing/reflection, hands-on skills practice, teamwork/communication training, skills maintenance/retention and examination purposes when patients are unavailable.

Perceptions of the value of SBT differed by experience. Healthcare workers with less experience were more likely to identify skills acquisition as an advantage of SBT (45, 59%,>10 years vs 64, 71%≤10 years, p<0.05). Healthcare workers with ≤10 years of experience were more likely to identify examination purposes when patients are unavailable (23, 30%>10 years vs 40, 44%≤10 years, p<0.05), whereas those with >10 years of experience identified debriefing/reflection (25, 33%>10 years vs 17, 19%≤10 years) as advantages of SBT. The perceived advantages of simulation also varied significantly by the profession of respondents (see table 4).

All respondents thought that SBT could be expanded beyond the current scope. Physicians were more likely to advocate for expanded use of simulation for continued practice after initial training (84, 88% physician vs 39, 63% nurses, p<0.001). They were also more likely to advocate for simulation for teaching (73, 76% physicians vs 16, 26% nurses, p<0.001) and research (65, 68% physicians vs 14, 23% nurses, p<0.001). If facilities were available, nearly all respondents (92, 98% physicians; 52, 96% nurses) would recommend the use of online simulation for their centre.

## DISCUSSION

Using data from a national survey of paediatric healthcare workers, we explored the access to and perceived utility of various simulation modalities in a resource-scarce setting. Our study found that many healthcare workers lack access to skills-based simulation labs for manikin-based training. The perceived challenges to establishing skills-based simulation labs were comparatively greater for respondents at government healthcare facilities with the greatest identified barriers being the lack of funding and access to equipment such as manikins. This is in contrast with the abundance of dedicated simulation facilities in high-income countries.[22–24] Dedicated spaces and equipment for SBT are only the first step, there is also a need to develop

**Table 4** Perceived advantages of simulation-based training vary by profession

| Advantages of simulation-based training | Physician, n (%) | Nurse, n (%) | P value |
|---|---|---|---|
| Skills acquisition | 83 (86) | 27 (44) | <0.001 |
| Provides feedback | 47 (49) | 11 (18) | <0.001 |
| Step down training | 48 (50) | 21 (34) | NS |
| Monitoring and evaluation | 47 (49) | 16 (26) | <0.01 |
| Debriefing/reflection | 34 (35) | 8 (13) | <0.01 |
| Hands-on skills practice | 64 (67) | 18 (29) | <0.001 |
| Teamwork/communication training | 55 (57) | 23 (37) | <0.05 |
| Skills maintenance/retention | 54 (56) | 15 (24) | <0.001 |
| Examination purposes when patients are unavailable | 56 (58) | 9 (15) | <0.001 |

locally relevant simulation cases and to train simulation instructors in the techniques of simulation facilitation and debriefing.[24 25]

The perceived utility of SBT may vary by profession and setting. Although many of our respondents identified specific ways in which SBT could be used, their responses varied by profession and experience. A variety of approaches have been described for interprofessional education including role play, manikin-based and virtual simulations. Interprofessional curricula may have differing impacts on learners of different professions.[26–28] Interprofessional virtual simulations have been shown to lead to varying changes in attitudes for students of different health professions.[28] It is therefore reasonable to infer that healthcare workers in different professions may benefit from SBT in different ways.

Healthcare workers were open to the expansion of simulation for teaching, continuing education and research and supported the introduction of online SBT. Online SBT is made more feasible than manikin-based simulation in resource-scarce settings by the widespread availability of mobile phones.[29] We confirmed a high percentage of smartphone use among healthcare workers in our study and low concern for potential barriers such as lack of internet access or inconsistent power supply. The integration of SBT into medical and nursing school curricula provides early exposure to SBT.[24] Establishing simulation programme at public and private healthcare facilities would enable the development of contextually appropriate simulation curricula and instructor courses in simulation facilitation, debriefing and research.[22 23]

A broad grass-roots approach that engages stakeholders in training institutions, state and national ministries of health, ministries of education, industry and health professional organisations is needed to support the integration of SBT into pre-service training and continuing education programme for in-service healthcare workers. Continuing education programme support the acquisition and retention of skills after initial training and have been important sources of sustainable funding for SBT in high-income settings.[7 13–16] These mechanisms may be leveraged to support SBT in resource-scarce settings.

This study had some limitations. This was a cross-sectional survey developed by the authors and was not a validated instrument. The data were obtained by self-report and could be subject to selection and recall bias. The survey was administered to attendees at a national paediatric conference. Although respondents worked at both training and non-training institutions and in both public and private settings, their attendance at the conference may indicate that they may be more likely to be supportive of academic pursuits, including SBT. Although physicians (both consultants and registrars) and nurses were represented in this study, other cadres of healthcare workers including community health extension workers and medical officers were not well represented and the utilisation of simulation in these groups could be a subject for future study.

## CONCLUSIONS

The access of healthcare workers to SBT is limited in resource-scarce settings. While acknowledging the challenges of lack of awareness, limited access to equipment and funding, respondents identified many areas in which SBT has utility including skills acquisition, hands-on skills practice and communication training. Healthcare workers were open to the use of online SBT and expressed the need to expand SBT beyond the current scope for pre-service and in-service health professional training in Nigeria.

**Acknowledgements** The authors would like to acknowledge the healthcare workers who participated in this study.

**Contributors** All authors have made substantial contributions to the planning, conduct, analyses and interpretation of findings and reporting of the work described in the article and have agreed to be accountable for all aspects of the work, its accuracy and integrity. RU: responsible for the overall content as guarantor; wrote the first draft of the manuscript and revised and amended it with input from all authors who also approved the final version to be published. RU and VCE: formulated the study objectives and survey. IBF, BNE and PA: assisted VCE with data collection. EC: assisted with data entry. CS: performed statistical analysis.

**Funding** This work was supported by the Bill and Melinda Gates Foundation, grant number OPP1169873.

**Competing interests**  None declared.

**Patient consent for publication**  Not required.

**Ethics approval**  Seattle Children's Hospital Institutional Review Board STUDY00000262; University of Lagos Health Research Ethics Committee CMUL/HREC/09/18/445.

**Provenance and peer review**  Not commissioned; externally peer reviewed.

**Data availability statement**  All data relevant to the study are included in the article or uploaded as supplementary information.

**Open access**  This is an open access article distributed in accordance with the Creative Commons Attribution 4.0 Unported (CC BY 4.0) license, which permits others to copy, redistribute, remix, transform and build upon this work for any purpose, provided the original work is properly cited, a link to the licence is given, and indication of whether changes were made. See: https://creativecommons.org/licenses/by/4.0/.

**ORCID iDs**
Rachel Umoren http://orcid.org/0000-0003-2356-9278
Beatrice N Ezenwa http://orcid.org/0000-0001-7437-3211

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
