## [Reviewer comments · BMJ Open]

ARTICLE DETAILS

TITLE (PROVISIONAL)	Perspectives on Simulation-based Training from Paediatric Healthcare Providers in Nigeria: A National Survey
AUTHORS	Umoren, Rachel; Ezeaka, V Chinyere; Fajolu, Ireti B; Ezenwa, Beatrice N; Akintan, Patricia; Chukwu, Emeka; Spiekerman, Chuck

VERSION 1 – REVIEW

REVIEWER	Gary Geis Cincinnati Children's Hospital Medical Center United States of America
REVIEW RETURNED	28-Oct-2019

GENERAL COMMENTS	ABSTRACT - page 2: Somewhere need to state that you are analyzing for differences between physicians and nurses. That does not come across until the results section. Line 9: This is your measure, not your setting. Line 17: Please write out response rate, as 1st time you use this abbreviation. Lines 19-22: Your measure is your survey. These lines are unnecessary as written, as provide same info as your objectives above. Lines 24-28: Likely do not need this demographic detail in the abstract, as it is not one of your primary outcomes. Line 40: Consider placing "be" between words should and equipped. INTRODUCTION - page 3: Line 15: Not sure Paediatric needs to be capitalized, will leave up to editor(s). Line 15: You use both simulation-based training and simulation-based education in the intro. Would recommend you select one and stick with that. You use SBT in the abstract, so likely that is the one to use. If so, can abbreviate SBT here. Lines 42-45: This sentence is different than the one in the abstract: "The objective of this study was to explore the access to and perceived utility of various simulation modalities by in-service healthcare providers in a low resource setting." I would suggest you keep them consistent. Line 45: Overall, introduction is well written - it develops what is simulation, why it is standardized resuscitation programs, how attempts have been made with low cost manikins in low resource settings, and where the current obstacles/problems are in accessing equipment and space in low resource settings. One things that is missing is a hypothesis - was there one in place when designing the outcomes and measures?
--

METHODS - pages 3-5:

Page 3, lines 48-52: Can you provide more detail on development and testing/piloting of the instrument itself. For example, what is piloted for clarity, length, ease of use, etc.? If so, was it revised? Were the questions adopted/adapted from other published measures, or were they created de novo? Was it delivered in English, and if so were all the providers primarily English-speaking as a primary language? This added information would help the reader better assess the validity and reliability of your tool.

Page 4, line 12: Where you discuss informed consent, this is confusing, as noted about it was exempt from human subjects review. Please clarify for reader.

Page 4, line 18: I would recommend providing the actual survey tool as an appendix or online supplemental data. This would provide the reader with a better conceptualization of and clarity surrounding the tool, and also allow the authors to limit the text here in the manuscript.

Page 4, lines 32-36: To my knowledge, none of these are offered in virtual reality currently. If they are, can you note which ones are VR options?

Page 5, line 8: Based on the description of the measure, it does not seem there were any open ended questions. If so, please include that information here and potentially explain why you chose not to include any qualitative or free text responses.

Page 5, line 11: As suggested in the abstract, I would recommend you specifically state here you are analyzing for differences between physicians and nurses

Page 5, line 17: Although you include the Strobe Statement, it might be prudent to further clarify here that no power calculation or sample size calculation was performed as you had a fixed sample size, i.e. providers attending the conference.

RESULTS - pages 5-10:

Page 5, line 20 - For clarity I would consider spelling out response rate, as this is not a common abbreviation (some might assume this is relative risk).

Page 5, Table 1 - it may be helpful for the reader to know what a registrar/house officer, medical officer and community health extension worker/officer are; specifically are these physicians?

Page 6, Table 1: Although listed above, it might be helpful to note in this table that North Central is the location where the conference was held. Also, what does FCT stand for?

Page 6, line 43: I would not say 37% is "most"; maybe restate as the range with the highest response was in the 5-10 year of experience.

Page 6, line 44: Again, although listed above, it might be helpful to note (if not revised in the table) here in the text that North Central is the location where the conference was held.

Page 7, lines 51-52: Neither profession or years in practice were statistically significant, do you want to word this sentence differently? Also, recommend not having See Table 3 as stand alone sentence, maybe instead put is after facility as (Table 3).

Page 8, line 4: I am not sure 7% versus 5%, with such low number, is actually statistically significant. Can you recheck the statistics?

Page 8, line 6: Consider substituting the word "and" for the comma before nurses.

Page 9, lines 5-6: Would clarify that there were significant difference in only two of the proposed questions/domains. Also, please not in the text that this question related to only skill-based

	simulation where differences were seen between private and governmental facilities. That info is in the Figure, but not apparent in the text. Page 9, lines 29-34: As written, the information appears misstated regarding "when patients are unavailable" as the sentence says workers with more than 10 years of experience were more likely, yet the numbers show 30% which is less than 44% in the <10 years group. Please clarify. Page 10, line 30: As written, the results section is very difficult for the reader to follow. There are too many numbers, i.e. 84, 88% physician vs. 39, 63% nurses, p<0.001. Maybe state early in the results section __ physicians responded and __ nurses responded. Then state, from here on out in the manuscript we will just report percentages. DISCUSSION - pages 10-11: This is where you should discuss how your results should be interpreted. Just stating your results over is not truly discussing them. I would recommend going back to the objectives of your study: The objective of this study was to explore the access to and perceived utility of various simulation modalities by in-service healthcare providers in a low resource setting. Can you write a paragraph or two discussing "access to" simulation, but not by repeating the results info. Please place your results in the context of your setting and population, explain reasons behind it, support it with other data outside of this survey and then propose possible options to improve the lack of access. Then, a 3rd and 4th paragraph could be focused on the perceived utility of simulation within your respondents. You do get to some of this information in the next two paragraphs. Page 11, line 25: This limitations section is underwritten. I would suggest using the published literature on survey-based instruments to further list and then discuss some/all the limitations to this methodology. For example, selection of your participants are a likely limitation as this was a convenience sample of providers who were actually attending a conference. One could argue those attendees are more likely supportive of academic pursuits, and thus more likely to support SBT and bias your results towards SBT. To discuss this, you need to argue for/against whether this cohort is representative of pediatric providers in the country. Page 11, lines 26-27: Please use additional text here to explain this for the reader; i.e. why would a predominantly female-based response underestimate exposure? CONCLUSION - page 11: Would recommend limiting your conclusions to the results you have obtained in your study. Most of this is opinion-based or at least not supported by your data.
--	---

REVIEWER	Aimee Sarti The Ottawa Hospital Canada
REVIEW RETURNED	05-Nov-2019

GENERAL COMMENTS	This study is an important contribution to the global simulation-based education domain. I enjoyed reading the manuscript, and thought the findings were not unexpected, it is clear that simulation-based education is a worthwhile goal for many African countries. The manuscript is well referenced with many interesting
---

	and current citations. My main concern with the paper is focused on the methods. In particular, there is virtually no discussion as to how the survey was developed (e.g., expert panel, etc.), nor is there any discussion if the survey was pilot tested and subsequently revised. In particular, it is unknown if the survey instrument was validated, and without appending the instrument to the manuscript there is little information as to the quality of the survey. Further, the use of Fisher's Exact test to examine the relationship between variables needs more explanation. That is, how were the calculations made (e.g., SPSS?). The manuscript requires additional details both in the methods as well as in providing some general context for the reader. Concerning the former, it is not clear why some categories were collapsed (see for example Table 2). It would help to provide some text as to why these decisions were made (even if they seem obvious to the authors). Finally, it would be helpful to provide some context as to the linkage between the researchers from Seattle to the Nigerian setting.
--	---

VERSION 1 – AUTHOR RESPONSE

Reviewer: 1

Reviewer Name: Gary Geis

Institution and Country: Cincinnati Children's Hospital Medical Center United States of America

Please state any competing interests or state 'None declared': None

Please leave your comments for the authors below ABSTRACT - page 2:

Somewhere need to state that you are analyzing for differences between physicians and nurses. That does not come across until the results section.

We examined the relationship between demographic characteristics and the respondents' access and exposure to simulation-based training facilities in their healthcare facility. Profession is specified as one of the demographic characteristics that was examined. Although space constraints make it difficult to expand on this further in the abstract, we have added additional details for clarity in the Data analysis section.

Line 9: This is your measure, not your setting.

The statement "A paper-based 25-item cross-sectional survey on simulation-based training (SBT) was administered to a convenience sample of healthcare workers from secondary and tertiary healthcare facilities." has been moved to the Primary and Secondary Outcome Measures section of the abstract.

Line 17: Please write out response rate, as 1st time you use this abbreviation.

"RR" has been changed to "response rate".

Lines 19-22: Your measure is your survey. These lines are unnecessary as written, as provide same info as your objectives above.

The Primary and Secondary Outcome Measures section has been edited to read:

"A paper-based 25-item cross-sectional survey on simulation-based training (SBT) was administered to a convenience sample of healthcare workers from secondary and tertiary healthcare facilities."

Lines 24-28: Likely do not need this demographic detail in the abstract, as it is not one of your primary outcomes.

Line 40: Consider placing "be" between words should and equipped.

Thank you. This sentence has been edited in line with the updated conclusion of the paper and now reads:

"The access of healthcare workers to SBT is limited in resource-scarce settings. While acknowledging the challenges, respondents identified many areas in which SBT may be useful, including skills acquisition, skills practice and communication training. Healthcare workers were open to the use of online SBT and expressed the need to expand SBT beyond the current scope for health professional training in Nigeria."

INTRODUCTION - page 3:

Line 15: Not sure Paediatric needs to be capitalized, will leave up to editor(s).

"Paediatric" has been edited to read "paediatric" throughout the document.

Line 15: You use both simulation-based training and simulation-based education in the intro. Would recommend you select one and stick with that. You use SBT in the abstract, so likely that is the one to use. If so, can abbreviate SBT here.

This change has been made to simulation-based training (SBT) consistently throughout the document.

Lines 42-45: This sentence is different than the one in the abstract: "The objective of this study was to explore the access to and perceived utility of various simulation modalities by in-service healthcare providers in a low resource setting." I would suggest you keep them consistent.

Thank you. The statement in the introduction has been updated to match the abstract.

Line 45: Overall, introduction is well written - it develops what is simulation, why it is standardized resuscitation programs, how attempts have been made with low cost manikins in low resource settings, and where the current obstacles/problems are in accessing equipment and space in low resource settings. One things that is missing is a hypothesis - was there one in place when designing the outcomes and measures?

Thank you. As this was a descriptive study, we did not formulate a specific hypothesis.

METHODS - pages 3-5:

Page 3, lines 48-52: Can you provide more detail on development and testing/piloting of the instrument itself. For example, what is piloted for clarity, length, ease of use, etc.? If so, was it revised? Were the questions adopted/adapted from other published measures, or were they created de novo? Was it delivered in English, and if so were all the providers primarily English-speaking as a primary language? This added information would help the reader better assess the validity and reliability of your tool.

Additional information on the development and testing of the instrument has been added to the methods section.

Page 4, line 12: Where you discuss informed consent, this is confusing, as noted about it was exempt from human subjects review. Please clarify for reader.

The statement was edited for clarity to indicate that information about the study was provided to the participants as part of the survey.

Page 4, line 18: I would recommend providing the actual survey tool as an appendix or online supplemental data. This would provide the reader with a better conceptualization of and clarity surrounding the tool, and also allow the authors to limit the text here in the manuscript.

The actual survey tool has been uploaded as a supplemental file.

Page 4, lines 32-36: To my knowledge, none of these are offered in virtual reality currently. If they are, can you note which ones are VR options?

Respondents were asked questions about their exposure to online paediatric simulations: Neonatal Resuscitation Program eSIM, HeartCode (PALS Online course), Online BLS, and Online ACLS course.

A clarifying statement has been added:

No examples of virtual reality simulations specific for paediatrics training were available at the time of the survey, but respondents were asked if they had ever used virtual reality simulations.

Page 5, line 8: Based on the description of the measure, it does not seem there were any open ended questions. If so, please include that information here and potentially explain why you chose not to include any qualitative or free text responses.

The survey included one open-ended question as a follow-up to the question: "If all facilities were available, would you recommend online simulation for your center?" Yes/No. If No, please state your reasons. Nearly all respondents (98%) responded "Yes" and there were no responses to this followup question.

Page 5, line 11: As suggested in the abstract, I would recommend you specifically state here you are analyzing for differences between physicians and nurses Page 5, line 17: Although you include the Strobe Statement, it might be prudent to further clarify here that no power calculation or sample size calculation was performed as you had a fixed sample size, i.e. providers attending the conference.

This statement has been clarified to read:

Data were analysed using descriptive statistics and the Fisher's Exact test to examine the relationship between demographic characteristics and respondents' access and exposure to SBT facilities in their institution or healthcare facility as well as their perceptions of the benefits and challenges in using SBT in their facility. We specifically compared the impact of demographic characteristics such as profession (physician or nurse), years in practice and type and location of practice; on access to SBT, perceived challenges of SBT and perceived utility of SBT.

An additional statement has been included:

No power calculation or sample size calculation was performed as the sample size was fixed, i.e. healthcare workers attending the conference.

RESULTS - pages 5-10:

Page 5, line 20 - For clarity I would consider spelling out response rate, as this is not a common abbreviation (some might assume this is relative risk).

For clarity, response rate has been spelt out.

Page 5, Table 1 - it may be helpful for the reader to know what a registrar/house officer, medical officer and community health extension worker/officer are; specifically are these physicians?

Physicians and non-physicians have been grouped separately in Table 1 for clarity.

Page 6, Table 1: Although listed above, it might be helpful to note in this table that North Central is the location where the conference was held. Also, what does FCT stand for?

This clarification has been included in the legend of Table 1.

Page 6, line 43: I would not say 37% is "most"; maybe restate as the range with the highest response was in the 5-10 year of experience.

The statement has been clarified to read:

Most respondents reported practicing for 10 years or less (90, 54%) and many practiced in a tertiary care setting (72, 43%).

Page 6, line 44: Again, although listed above, it might be helpful to note (if not revised in the table) here in the text that North Central is the location where the conference was held.

This clarification has been included in the legend of Table 1.

Page 7, lines 51-52: Neither profession or years in practice were statistically significant, do you want to word this sentence differently? Also, recommend not having See Table 3 as stand alone sentence, maybe instead put is after facility as (Table 3).

Thank you. This statement has been reworded as follows:

Exposure to manikin-based training varied significantly by type of facility and geographic location (Table 3).

Page 8, line 4: I am not sure 7% versus 5%, with such low number, is actually statistically significant. Can you recheck the statistics?

Thank you. The statement has been clarified as follows:

Physicians were the group most likely to have been exposed to manikin-based paediatric training programs such as Helping Babies Breathe (29, 30% physicians vs. 12, 19% nurses vs. 1, 11% community health workers, $p < 0.001$) or online training in neonatal resuscitation using the NRP eSIM (7, 7% physician vs. 3, 5% nurses vs. 0, 0% community healthcare workers, $p < 0.05$).

Page 8, line 6: Consider substituting the word "and" for the comma before nurses.

This substitution has been made.

Page 9, lines 5-6: Would clarify that there were significant difference in only two of the proposed questions/domains. Also, please not in the text that this question related to only skill-based simulation where differences were seen between private and governmental facilities. That info is in the Figure, but not apparent in the text.

This information has been included in the reworded statement:

Lack of curriculum and lack of funding were perceived as less of a barrier to establishing a skills-based simulation lab by respondents from private healthcare facilities compared with respondents from government facilities ($p < 0.05$) (Figure 1).

Page 9, lines 29-34: As written, the information appears misstated regarding "when patients are unavailable" as the sentence says workers with more than 10 years of experience were more likely, yet the numbers show 30% which is less than 44% in the <10 years group. Please clarify.

Thank you. The statement has been clarified as follows:

Healthcare workers with less than or equal to 10 years of experience were more likely to identify examination purposes when patients are unavailable (23, 30% > 10 years vs. 40, 44% \leq 10 years, $p < 0.05$), while those with more than 10 years of experience identified debriefing/reflection (25, 33% > 10 years vs. 17, 19% \leq 10 years) as advantages of SBT.

Page 10, line 30: As written, the results section is very difficult for the reader to follow. There are too many numbers, i.e. 84, 88% physician vs. 39, 63% nurses, $p < 0.001$. Maybe state early in the results section __ physicians responded and __ nurses responded. Then state, from here on out in the manuscript we will just report percentages.

The statement has been reworded for clarity and now reads:

Physicians were more likely to advocate for expanded use of simulation for continued practice after initial training (84, 88% physician vs. 39, 63% nurses, $p < 0.001$). They were also more likely to advocate for simulation for teaching (73, 76% physicians vs. 16, 26% nurses, $p < 0.001$), and research (65, 68% physicians vs. 14, 23% nurses, $p < 0.001$).

DISCUSSION - pages 10-11:

This is where you should discuss how your results should be interpreted. Just stating your results over is not truly discussing them. I would recommend going back to the objectives of your study: The

objective of this study was to explore the access to and perceived utility of various simulation modalities by in-service healthcare providers in a low resource setting. Can you write a paragraph or two discussing "access to" simulation, but not by repeating the results info. Please place your results in the context of your setting and population, explain reasons behind it, support it with other data outside of this survey and then propose possible options to improve the lack of access. Then, a 3rd and 4th paragraph could be focused on the perceived utility of simulation within your respondents. You do get to some of this information in the next two paragraphs.

Thank you for your suggestions. The objective of the study has been referenced in the first paragraph of the discussion which has been edited extensively in line with the suggestions.

Page 11, line 25: This limitations section is underwritten. I would suggest using the published literature on survey-based instruments to further list and then discuss some/all the limitations to this methodology. For example, selection of your participants are a likely limitation as this was a convenience sample of providers who were actually attending a conference. One could argue those attendees are more likely supportive of academic pursuits, and thus more likely to support SBT and bias your results towards SBT. To discuss this, you need to argue for/against whether this cohort is representative of pediatric providers in the country.

Thank you. This statement has been added to the limitation:

The survey was administered to attendees at a national paediatric conference. Although respondents worked at both training and non-training institutions and in both public and private settings, their attendance at the conference may indicate that they may be more likely to be supportive of academic pursuits, including SBT.

Page 11, lines 26-27: Please use additional text here to explain this for the reader; i.e. why would a predominantly female-based response underestimate exposure?

A review of recent literature shows this text was unsupported and has been removed from the limitations.

CONCLUSION - page 11:

Would recommend limiting your conclusions to the results you have obtained in your study. Most of this is opinion-based or at least not supported by your data.

The conclusion has been reworded to state:

The access of healthcare workers to SBT is limited in resource-scarce settings. However, respondents identified many areas in which SBT has utility including skills acquisition, hands-on skills practice and communication training. Lack of awareness, access to equipment and funding were identified as challenges to SBT. Healthcare workers were open to the use of online SBT and expressed the need to expand SBT beyond the current scope for pre-service and in-service health professional training in Nigeria.

Reviewer: 2

Reviewer Name: Aimee Sarti

Institution and Country: The Ottawa Hospital Canada

Please state any competing interests or state 'None declared': None declared

Please leave your comments for the authors below This study is an important contribution to the global simulation-based education domain. I enjoyed reading the manuscript, and thought the findings were not unexpected, it is clear that simulation-based education is a worthwhile goal for many African countries. The manuscript is well referenced with many interesting and current citations.

My main concern with the paper is focused on the methods. In particular, there is virtually no discussion as to how the survey was developed (e.g., expert panel, etc.), nor is there any discussion if the survey was pilot tested and subsequently revised. In particular, it is unknown if the survey

instrument was validated, and without appending the instrument to the manuscript there is little information as to the quality of the survey.

Additional information on the development and testing of the instrument has been added to the methods section. The actual survey tool has been uploaded as a supplemental file.

Further, the use of Fisher's Exact test to examine the relationship between variables needs more explanation. That is, how were the calculations made (e.g., SPSS?).

This information has been included in the Data analysis section which now reads:
SAS 9.4 software [SAS Institute, Cary NC] was used for the analysis.

The manuscript requires additional details both in the methods as well as in providing some general context for the reader. Concerning the former, it is not clear why some categories were collapsed (see for example Table 2). It would help to provide some text as to why these decisions were made (even if they seem obvious to the authors).

We have added a statement to the data analysis section to clarify that some categories were collapsed for analysis:

In some cases, subcategories of profession (e.g. physician types included consultant physician, registrar, house officer, medical officer), years in practice and geographic location (North vs. South geopolitical zones) were collapsed for comparison due to small numbers of responses in individual categories.

Finally, it would be helpful to provide some context as to the linkage between the researchers from Seattle to the Nigerian setting.

The lead researchers are collaborators in simulation research in Nigeria and this study was undertaken to investigate the current state of simulation-based training in Nigeria. This context is provided in the methods section which now reads:

A 35-item cross-sectional survey was created by the investigators (RU, CE) who are simulation research collaborators from the University of Washington/Seattle Children's Hospital and the University of Lagos with questions on access to SBT facilities and perceptions on SBT in paediatric settings. Input was obtained from experienced simulation educators and healthcare professionals practicing in the U.S. and in Nigeria. The survey was piloted for clarity and ease of use among Nigerian paediatric healthcare professionals and revised based on feedback. The survey was designed to be delivered in English and intended for administration to paediatric professionals.

VERSION 2 – REVIEW

REVIEWER	Gary Geis Cincinnati Children's Hospital Medical Center Cincinnati, OH United States of America
REVIEW RETURNED	30-Dec-2019

GENERAL COMMENTS	Thank you for allowing me to review this resubmission. I feel you have been very responsive to my original review and I feel the submission is now acceptable for publication, with only a few minor comments as follows: Abstract: Page 2, line 20 - please add "of age" after 31-40 years Introduction: No comments Methods:
--

	Page 3, line 48 - Thank you for including this instrument in your resubmission. I would recommend that you reference it here in the text as a supplemental file, instead of on line 3 of the next page. Page 4, line 18 - Please remove second period at the end of the sentence Page 4, line 24, Measures sub-section - As you have now attached the measure (survey) as supplemental content, I think you could eliminate a lot of the text in this measures section. My suggestion would be to just list your domains. For example: "Our 25-item survey based measure inquired about four main domains of simulation-based training: access to SBT facilities, exposure to SBT, challenges to SBT, and perceptions of SBT. Within the exposure domain, no examples of virtual reality simulations specific to pediatric training were available at the time of the survey, but respondents were asked if they had ever used virtual reality simulation." Results: No comments Discussion: No comments Conclusion: Page 15, line 15, Conclusion section - I would consider making the conclusion here mirror the one in the abstract, which will provide consistency for the reader.
--	--

REVIEWER	Aimee Sarti Department of Critical Care, The Ottawa Hospital Canada
REVIEW RETURNED	11-Dec-2019

GENERAL COMMENTS	I am satisfied that the authors have successfully addressed all of my comments during the first round of revisions. That being said, however, I do think it is important that the authors included that their survey is not a validated instrument in the limitations section of the manuscript.
--

VERSION 2 – AUTHOR RESPONSE

Reviewer: 1

Reviewer Name: Gary Geis

Institution and Country:

Cincinnati Children's Hospital Medical Center

Cincinnati, OH

United States of America

Please state any competing interests or state 'None declared': None declared

Please leave your comments for the authors below

Thank you for allowing me to review this resubmission. I feel you have been very responsive to my original review and I feel the submission is now acceptable for publication, with only a few minor comments as follows:

Abstract:

Page 2, line 20 - please add "of age" after 31-40 years

This revision has been made.

Introduction: No comments

Methods:

Page 3, line 48 - Thank you for including this instrument in your resubmission. I would recommend that you reference it here in the text as a supplemental file, instead of on line 3 of the next page. This revision has been made.

Page 4, line 18 - Please remove second period at the end of the sentence

This revision has been made.

Page 4, line 24, Measures sub-section - As you have now attached the measure (survey) as supplemental content, I think you could eliminate a lot of the text in this measures section. My suggestion would be to just list your domains. For example:

"Our 25-item survey based measure inquired about four main domains of simulation-based training: access to SBT facilities, exposure to SBT, challenges to SBT, and perceptions of SBT. Within the exposure domain, no examples of virtual reality simulations specific to pediatric training were available at the time of the survey, but respondents were asked if they had ever used virtual reality simulation."

This section has been revised as suggested to remove the text on response options.

Results: No comments

Discussion: No comments

Conclusion:

Page 15, line 15, Conclusion section - I would consider making the conclusion here mirror the one in the abstract, which will provide consistency for the reader.

Thank you. The conclusion has been revised to read:

"The access of healthcare workers to SBT is limited in resource-scarce settings. While acknowledging the challenges of lack of awareness, limited access to equipment and funding, respondents identified many areas in which SBT has utility including skills acquisition, hands-on skills practice and communication training. Healthcare workers were open to the use of online SBT and expressed the need to expand SBT beyond the current scope for pre-service and in-service health professional training in Nigeria."

Reviewer: 2

Reviewer Name: Aimee Sarti

Institution and Country:

Department of Critical Care, The Ottawa Hospital
Canada

Please state any competing interests or state 'None declared': None declared

Please leave your comments for the authors below

I am satisfied that the authors have successfully addressed all of my comments during the first round of revisions. That being said, however, I do think it is important that the authors included that their survey is not a validated instrument in the limitations section of the manuscript.

This has been added to the limitations section.